# Effects of Hyperthermia and Hyperthermic Intraperitoneal Chemoperfusion on the Peritoneal and Tumor Immune Contexture

**DOI:** 10.3390/cancers15174314

**Published:** 2023-08-29

**Authors:** Daryl K. A. Chia, Jesse Demuytere, Sam Ernst, Hooman Salavati, Wim Ceelen

**Affiliations:** 1Department of Surgery, National University Hospital, National University Health System, Singapore 119074, Singapore; 2Department of Human Structure and Repair, Experimental Surgery Lab, Ghent University, 9052 Ghent, Belgium; jesse.demuytere@ugent.be (J.D.); sam.ernst@ugent.be (S.E.); hooman.salavati@ugent.be (H.S.); 3Cancer Research Institute Ghent, 9000 Ghent, Belgium; 4Department of GI Surgery, Ghent University Hospital, 9000 Ghent, Belgium

**Keywords:** hyperthermia, peritoneal carcinomatosis, peritoneal metastases, intraperitoneal chemotherapy

## Abstract

**Simple Summary:**

Cancer that spreads to the lining of the abdomen, the peritoneum, is currently treated with heated chemotherapy and surgery. However, the effects of the high temperature on the cancer cells and the immune cells are still unclear. In this review, we summarize the available data to show that high temperatures may have both positive and negative effects, and that further study is necessary to distinguish these effects.

**Abstract:**

Hyperthermia combined with intraperitoneal (IP) drug delivery is increasingly used in the treatment of peritoneal metastases (PM). Hyperthermia enhances tumor perfusion and increases drug penetration after IP delivery. The peritoneum is increasingly recognized as an immune-privileged organ with its own distinct immune microenvironment. Here, we review the immune landscape of the healthy peritoneal cavity and immune contexture of peritoneal metastases. Next, we review the potential benefits and unwanted tumor-promoting effects of hyperthermia and the associated heat shock response on the tumor immune microenvironment. We highlight the potential modulating effect of hyperthermia on the biomechanical properties of tumor tissue and the consequences for immune cell infiltration. Data from translational and clinical studies are reviewed. We conclude that (mild) hyperthermia and HIPEC have the potential to enhance antitumor immunity, but detailed further studies are required to distinguish beneficial from tumor-promoting effects.

## 1. Introduction

Peritoneal metastases (PM) commonly arise from gastrointestinal, appendiceal and gynecological malignancies and are inadequately treated by systemic chemotherapy alone due to the peritoneal-plasma barrier and poor blood supply of PM [1]. However, this peritoneal barrier was also recognized as a potential therapeutic strategy to exploit by delivering intraperitoneal (IP) therapy, particularly drugs that remain sequestered within the peritoneum while limiting systemic toxicity. This was demonstrated in an early randomized trial by Sugarbaker et al. showing the superiority of IP over systemic chemotherapy for advanced colorectal cancer in 1985 [2]. The addition of hyperthermia to IP chemotherapy has shown to be feasible since the 1980s, and thought to have direct anti-tumor cytotoxic effects, synergize with chemotherapeutic agents and increase the depth of tissue penetration, becoming mainstream in the management of PM, collectively known as hyperthermic intraperitoneal chemotherapy (HIPEC) [3,4,5,6]. However, studies evaluating the impact of hyperthermia alone are lacking; while shown to be safe, few studies have evaluated the direct benefit of hyperthermic IP chemotherapy over normothermia for PM [7,8]. Studies investigating hyperthermic vs. normothermic IP chemotherapy for PM have found no difference in local tissue concentration of chemotherapy, apoptosis rates or pathological response in the hyperthermic group, failing to substantiate the assumptions regarding the benefits of hyperthermia [8].

In current practice, the role of hyperthermia remains contentious in the treatment of PM. RCTs investigating the use of heated IP chemotherapy in gastric and colorectal PM have failed to demonstrate improvement in overall survival [9,10]. In epithelial ovarian cancer, RCTs have reported mixed results with two European RCTs reporting superior OS in patients undergoing CRS + HIPEC compared with CRS alone [11,12]. These findings were unfortunately not corroborated in a recent Korean RCT, which reported superior OS in patients undergoing interval CRS + HIPEC but not primary resection [13].

While the benefits of hyperthermia remain to be proven, concurrent negative effects on the peritoneum and its immune microenvironment may be possible. For instance, hyperthermia has been reported to impair cytolytic activity of cytotoxic T lymphocytes and impair the functions and counts of key immune cells including NK cells and monocytes [14,15]. Given the increasing role of immunotherapy in treating metastatic cancer, this may have deleterious effects on survival outcomes that remain poorly understood. With the increasing role of hyperthermia in novel peritoneal-directed therapies like pressurized intra-peritoneal aerosol chemotherapy (PIPAC) and nanoparticle-bound chemotherapeutic agents, there is an urgent need to better understand the advantages and limitations of hyperthermia in IP treatment for PM [16,17,18]. In this review, we outline the relevant peritoneal anatomy and immune environment, mechanisms of PM and the effects of hyperthermia on both tumor and normal peritoneum.

## 2. Immune Contexture of the Healthy Peritoneal Cavity

The peritoneal cavity holds the abdominal viscera and is lined by the peritoneum, a one-cell-layer-thick serous membrane of mesothelial cells (MC) with an underlying extracellular matrix. Anatomically, the peritoneum can be subdivided into the parietal peritoneum, which lines the abdominal wall, and the visceral peritoneum covering the viscera [19]. The MCs provide a smooth non-adhesive surface for the viscera by the production of peritoneal fluid (PF), surfactant, proteoglycans and glycosaminoglycans. Also, MCs facilitate transport of fluid and cells across the peritoneal cavity, form a physical barrier for pathogens and cancer cells, and engage a response to inflammation, tissue damage, and malignancy [20,21]. Mesothelial cells present antigens to CD4+ T cells, and express pattern recognition receptors (PRRs) that can recognize pathogens and damage-associated molecules [20]. Combined with their capacity to secrete cytokines, chemokines, growth factors, and extracellular matrix (ECM) components, they are essential in communication, tissue repair, and activation of the immune system in response to threats within the abdominal cavity. An important means for this communication is the PF, which contains both a molecular and cellular fraction and as such creates a peritoneal ecosystem across the abdominal cavity.

Throughout the peritoneal cavity, the peritoneum is fenestrated by stomata, openings in the mesothelial serous membrane connected to the lymphatic system. The main function of these stomata is to filter the PF and drain it to the lymph nodes [22,23]. Stomata located on the omentum and mesentery that are associated with fibroblasts and clusters of immune cells are termed milky spots or fat-associated lymphoid clusters (FALCs) and are of importance for immune cell turnover in the peritoneal cavity [23]. These tertiary lymphoid clusters are rich in both innate and adaptive immune cells, as well as lymphoid and myeloid cells, and have as main functions the maintenance of homeostasis, defense against threats, and tissue repair [23,24]. Very commonly found in these FALCs are neutrophils, a type of granulocyte capable of capturing circulating pathogens through phagocytosis and the use of their neutrophil extracellular traps (NETs). Upon inflammation, neutrophils can be recruited very promptly by the MC expression of the neutrophil recruitment chemokine, CXCL1, to help surmount the inflammatory cause [25]. Other phagocyting innate immune cells highly abundant in the peritoneal cavity are macrophages. Mice studies have shown the existence of two main macrophage populations in the peritoneum: the tissue resident GATA6-dependent large peritoneal macrophages (LPM) and the monocyte-derived IRF4-dependent small peritoneal macrophages (SPM) [26,27,28]. Whether these distinct peritoneal macrophage populations are also present in humans is yet to be confirmed.

One of the most common peritoneal immune cell types in the adaptive immune compartment are B cells. Specifically, the B1-cell subset, an innate-like B-cell type, rapidly produces high levels of IgM antibodies as a first response to peritoneal threats [29]. These B1 cells are also potent antigen presenting cells capable of activating CD4+ T cells. Among the T-cell population, both CD4+ T helper cells and CD8+ cytotoxic T cells prevail in the peritoneal cavity (Figure 1) [24].

## 3. Immune Contexture of Peritoneal Metastases

In peritoneal metastases (PM), a majority of the peritoneal immune microenvironment exhibits immunosuppressive phenotypes (e.g., regulatory T cells, exhausted T cells, tumor-associated macrophages and MDSCs) and thus support cancer growth [31]. The pathophysiology of PM can be described as a stepwise process, beginning with the detachment of cells from the primary tumor, transcoelomic transport with the peritoneal fluid, and attachment to the peritoneal surface or the underlying stroma. Subsequently, invasion into the subperitoneal space, proliferation of tumor cells, and angiogenesis lead to PM [31,32]. While these distinctive steps are part of a process shared by all abdominal malignancies metastasizing to the peritoneum, different cancer types may use unique mechanisms [32,33]. Moreover, PM can arise from primary tumors outside of the abdominal cavity such as melanoma, lung cancer, or lobular breast cancer, indicating a systemic route resulting in PM [34].

Recent insights suggest the existence of a pre-metastatic niche that precedes the formation of PM [31,33]. The peritoneal microenvironment is primed for PM by soluble tumor-secreted factors including cytokines such as TGF-β, VEGF, and exosomes [35,36]. As part of this process, the continuity of mesothelial cells is disrupted, and mesothelial-to-mesenchymal transition (MMT) occurs [37,38,39]. In addition, certain components of the peritoneal microenvironment such as macrophages and other immune cells are reprogrammed to allow tumor cells to establish as PM [31,33,40]. Integrating these concepts has resulted in a deeper understanding of the hallmarks of PM—namely paracrine factors, tumor-related factors, biomechanical forces, and the peritoneal microenvironment—that are intertwined in the formation of PM [41].

## 4. Non-Immune Related Effects of HIPEC

### 4.1. Effects on Peritoneal Blood Flow and Pharmacokinetics

Under physiological conditions, the estimated magnitude of the (visceral) peritoneal blood flow is 60–100 mL/min, representing 1–2% of the cardiac output [42]. Heat stress causes reflex vasodilation of skin vessels, promoting convective heat transfer from the body core to the skin. It is not known whether peritoneal microvessels exhibit a similar temperature-dependent vasodilation. However, since HIPEC causes an increased cardiac output and a lowered peripheral vascular resistance, it is likely that peritoneal blood flow increases with (mild) hyperthermia [43]. Due to vasodilation, hyperthermia may increase systemic uptake of chemotherapy, potentiating side effects. However, experimental results concerning the effect of hyperthermia on IP pharmacokinetics are inconclusive [44]. In a porcine oxaliplatin HIPEC model, tissue concentrations of platinum were increased in response to hyperthermia, with no increase in systemic absorption [45]. Other pharmacokinetic data in small animal models are conflicting, with some authors describing no significant effect of hyperthermia [46,47,48], while others demonstrated higher tissue platinum concentrations and decreased systemic absorption, which would indicate increased drug uptake [49,50]. However, pharmacokinetic analyses in animal models as reported by Sørensen et al. showed no effect of hyperthermia on MMC pharmacokinetics [51], while Klaver et al. did not demonstrate a survival benefit for HIPEC with MMC compared to normothermic chemoperfusion [52]. In an in vivo analysis of 50 patients, Xie et al. reported only a slight increase in plasma peak concentration of cisplatin when administered at 41 °C, with no increase in adverse events [53]. Multiple pre-clinical animal studies have demonstrated the increased depth of tissue penetration resulting from the addition of heat to IP chemotherapy [54,55,56,57]. This is explained by the temperature dependence of the diffusion process:(1)$$J=D_{0}e^{\frac{-Q_{d}}{RT}}\frac{dC}{dx}$$

With *J* the diffusion flux (mass transfer per unit area per unit time), *D* the diffusion coefficient (m^2^/s), *Q_d_* the activation energy (J/mol), *R* the gas constant, *T* the absolute temperature, and *dC*/*dx* the concentration gradient.

### 4.2. Effect on Chemotherapy Cytotoxicity

The three most common chemotherapeutic agents used in the context for HIPEC are mitomycin C (MMC), and the platinum agents cisplatin and oxaliplatin. All three of these agents have been described as being “thermally enhanced”, meaning their efficacy is increased by hyperthermic conditions [58]. However, most of these results are derived from in vitro or animal studies.

MMC combined with hyperthermia demonstrated increased cell-killing capacity of cancer cell lines in vitro [59,60], while a recent study by Helderman et al. disputes this [61]. Oxaliplatin and cisplatin efficacy has been described as augmented by hyperthermia, with increased apoptosis, drug uptake, and DNA damage reported in vitro in several colorectal and ovarian cancer cell lines, but these effects might be cell-line dependent [44,49,61,62,63].

### 4.3. Effects on Peritoneal and Gut Integrity

At temperatures higher than 43 °C, the risk of scald injury to the peritoneal surface increases [64]. Normal tissues in the vicinity of cancers are susceptible to thermal injury induced by hyperthermic IP treatment as has been verified by in vitro and in vivo studies [65]. In animal experiments exploring thermal dosimetry, significant histopathological damage to the bowel (rabbit, pig) was noted with exposure to 43 °C during 21–40 min, while liver injury was noted in rabbits from 41 min onward [65]. Pavel et al. reported small bowel damage induced by ≤20 mins of exposure to hyperthermia at 43 °C [66,67]. In animal models investigating intestinal injury, intestinal epithelial injury was seen at temperatures as low as 41.5 °C [67]. In addition, hyperthermia may affect the integrity of the mucosal intestinal barrier function. In animal studies, heat stress damaged mechanical and mucosal immune gut barriers reduces immune function of the intestinal mucosa and mesenteric lymphoid tissues, and leads to bacterial translocation [68,69,70]. Also, changes in core temperature have been linked to altered microbiome composition and function, and the effect of HIPEC on the integrity and function of the gut microbiome is an important question for future research [71].

## 5. Immune Effects of HIPEC

The potential immune stimulating effects of hyperthermia have been recognized since the observations of spontaneous tumor regression in patients with persistent fever. Endogenous hyperthermia (fever) stimulates the innate and adaptive immune system through multiple pathways, with interleukin-6 (IL-6) playing a central role [72]. The effects of hyperthermia on the peritoneal immune environment have not been studied in detail, and the mechanisms described below are not specific for HIPEC.

Hyperthermia alters the expression of thousands of genes involved in protein folding, cell cycle, mitosis, and cell death [73]. A prominent master regulator of the heat shock response is heat shock factor 1 (HSF1), which affects various cellular responses associated with tumorigenesis, including alteration of the tumor microenvironment, genome repair, and regulation of several cell death pathways including autophagy and ferroptosis [74]. Together with hypoxia-inducible factor 1 (HIF-1) [12], matrix metalloproteinase 3 (MMP-3), and heterochromatin protein 1, HSF1 triggers the production of heat shock proteins (HSPs) [75]. Heat shock proteins are molecular chaperones which protect intracellular proteins from misfolding or aggregation, inhibit cell death signaling cascades, and preserve essential intracellular signaling pathways. According to their molecular weight, they are classified as HSP110 (HSPH), HSP90 (HSPC), HSP70 (HSPA), HSP60 (HSPD), HSP40 (DNAJ), and small HSPs (HSPB). Although originally described in the setting of heat shock, they are known to be constitutively expressed and subject to stimulated expression by other stressors including cold. Overexpression of HSPs is common in a variety of solid cancer types, and they have been implicated in several oncogenic pathways. However, the effects seem to be tumor-type dependent, and efforts to develop HSP targeted therapies in clinical trials were unsuccessful [76]. Similarly, the effects of hyperthermia and HSPs on cancer immune surveillance are double edged. On the one hand, HSPs can bind tumor antigens and form complexes recognized by monocytes, macrophages, B cells, and dendritic cells, leading to cytotoxic T-cell activation [76]. On the other hand, HSF1 was shown to enhance the expression of PD-L1 in breast cancer. Additionally, compared to intra-cellular HSPs that are released by normal cells in response to trauma or stress states, HSPs are also transported into the extracellular compartment by non-canonical secretory pathways and may play a crucial role in promoting and sustaining key hallmarks of peritoneal carcinogenesis [77,78,79]. For instance, extracellular HSPs (eHSPs) have been shown to promote ECM remodeling to increase production of collagen and fibronectin production as scaffolding for tumor expansion [77]. eHSPs also contribute to stromal cell activation such as cancer-associated fibroblasts, which further secrete cytokines to maintain a pro-tumor environment [77,80]. eHSPs have also been shown to protect cancer cells from apoptosis or cell death by a variety of differing pathways [81]. eHSPs also directly interact with cancer cells via the paracrine effect on surface receptors such as the EGF receptor family, toll-like receptors and lipoprotein receptor-related proteins, which are implicated in cancer proliferation, apoptosis inhibition, and cancer progression [82] (Figure 2).

Of note, the effects are dependent on temperature: thermal ablation methods including microwave ablation result in the release of neoantigens and DAMPs, and provide the rationale for ongoing clinical studies that combine local ablation with immune checkpoint inhibitors [83]. Clearly, these mechanisms do not apply to the mild hyperthermia used during HIPEC.

Mild hyperthermia enhances immune cell recruitment by stimulating vascular perfusion and upregulation of vascular adhesion molecules including ICAM-1, which promote immune cell infiltration [84]. Several preclinical studies showed that mild hyperthermia results in reduced tumor growth and improved animal survival, in parallel with enhanced infiltration and activation of immune cell populations including natural killer (NK), CD4+ T, and CD8+ T cells. As an example, Toraya-Brown et al. found that treatment of murine melanoma with mild hyperthermia (43 °C during 30 min) using iron oxide nanoparticles combined with an alternating magnetic field resulted in activated dendritic cells (DCs) and subsequently CD8+ T cells in the draining lymph node, and conferred resistance against rechallenge [85]. However, these effects were not observed when using higher temperatures (45 °C).

Several authors have investigated the effects of HIPEC on tumor immune cell infiltration in animal models. Nevo et al. found that HIPEC with mitomycin C led to increased infiltration of CD4+, CD8+, CD68+, and CD20+ cells into omental and visceral metastases in a mouse model of colorectal PM (MC38/C57BL) [86]. In the same model, the authors showed that combining HIPEC with anti-PD-1 treatment resulted in improved survival compared with HIPEC alone [87]. Wu et al. administered IP chemohyperthermia in a mouse ovarian cancer model, and observed that the addition of hyperthermia restored the number of intraperitoneal macrophages and dendritic cells, which were depleted in the chemotherapy alone group [87]. Zunino et al. performed vaccination experiments in a CT26/Balb/c murine colorectal cancer model [88]. They found that after vaccination with mitomycin C or hyperthermia exposed CT26 cells, tumor growth was inhibited after injection of CT26 cells in the opposite flank. Mechanistically, they showed that the antitumor vaccination was mediated by HSP90 exposure on the cell surface of the dying cells.

### Clinical and Translational Studies

Published data regarding the clinical benefits of hyperthermia in the context of PM has thus far been limited and inconsistent. One clinical trial of prophylactic HIPEC in gastric cancer was performed with mitomycin C (MMC) and oxaliplatin at normothermic or hyperthermic conditions, demonstrating a survival benefit for the hyperthermia group [89]. However, these findings were not confirmed in a larger meta-analysis comparing hyperthermic vs. normothermic IP chemotherapy for gastric cancer patients with PM [90]. The landmark CYTO-CHIP study evaluating a retrospective cohort of patients with PM from gastric cancer demonstrated a survival benefit with the addition of HIPEC to cytoreductive surgery, although it is unclear whether the benefit arose from hyperthermia, IP chemotherapy, or a synergistic combination of both factors [91]. Similarly, the RCT by Casado-Adam et al. evaluated patients with ovarian cancer undergoing normothermic vs. hyperthermic IP paclitaxel. In addition to finding no significant increase in local tissue concentrations of chemotherapy in patients receiving heated treatments, markers of apoptosis (caspase-3), pathological response, and cellular proliferation (p53, p27, p21, Ki67) were also not significantly different between normothermic and hyperthermic treatments, casting doubt on the purported benefits of adding heat to IP treatment [8]. To address some of these gaps in current knowledge, a number of ongoing RCTs aim to re-examine fundamentals of hyperthermic IP therapy. For instance, the OvIP1 trial (NCT02567253) aims to investigate the efficacy of heated vs. normothermic IP cisplatin in patients with ovarian cancer [92]. Ongoing and completed RCTs are summarized in Table 1.

While the clinical benefit of hyperthermia for PM treatment has yet to be demonstrated, several translational studies have alluded to a biological basis for hyperthermic treatment at a molecular level. Dellinger et al. examined pre- and post-treatment samples from patients with ovarian cancer undergoing HIPEC and reported heat shock proteins as one of the most upregulated genes after treatment. In addition, an increase in programmed cell death protein 1 expression on CD8+ T cells and downregulated DNA repair pathways within cancer islands were associated with significantly increased progression-free survival after hyperthermic IP treatment and could serve as a predictive biomarker in the future [93]. In the same study, the gene expression changes seen post-HIPEC were similar in tumor and normal tissue, albeit with a decreased magnitude of change seen in normal tissues. Metabolic pathways were downregulated and immune pathways were upregulated to a greater extent in tumor compared to normal tissue as well, suggesting that hyperthermia enabled a certain specificity for targeted treatment of the tumor [93]. This was further validated with post-HIPEC PM tumors demonstrating significant apoptosis and inflammatory change on immunohistochemistry while normal tissue only exhibited the latter [93]. Similarly, Moukarzel et al. examined transcriptomic changes after HIPEC in ovarian tumors and normal tissues, and demonstrated differential gene expression between tumors (heat response- and protein folding-related genes) and normal tissues where genes relating to immune response were altered [94]. A difference in upregulated HSPs was noted between tumors and normal tissue, with an overall increase of HSP90, GRP75, and Ubiquitin+1 seen in normal tissues compared to a decrease in tumor tissues; an increase in HSP27, HSP32, HSP40, HSP60 protein expression levels were observed in tumor tissues as compared to normal tissue and only HSP70 expression was increased in both normal and tumor tissue after HIPEC exposure [94]. However, the consequence of these changes in transcriptome are not apparent. In colorectal and gastrointestinal related PM, exposure to HIPEC resulted in upregulation of intracellular heat shock proteins (HSP) and resultant amplification of proliferation markers and antiapoptotic proteins, as well as chemoresistance against fluoropyrimidine, MMC, and oxaliplatin agents (Figure 3) [77,82,95,96,97]. A recent meta-analysis of published transcriptomic changes in human cancer cells undergoing heat stress evaluated a total of 18 datasets (2 unpublished) and identified considerable inter-study variability, as well as an apparent absence of a “universal” heat stress response, highlighting the need for further in-depth collaborative efforts to standardize clinical and experimental protocols crucial to understanding the response of PM to hyperthermia [73].

Lastly, the impact of hyperthermia on the resident immune microenvironment in the peritoneum needs further evaluation in the era of immunotherapy. Moukarzel et al. reported similar immune cell compositions between normal tissue and ovarian tumors, with normal tissue demonstrating increased CD8 T cells, activated and resting mast cells, activated NK cells and neutrophils while no change in immune cell composition was noted in tumor samples before and after HIPEC [73]. This suggests that there was no significant impact of hyperthermic IP chemotherapy on the proportions of immune cells in PM. In contrast, Dellinger et al. reported a significant increase in % PD-1 expression in CD8+ tumor-infiltrating lymphocytes and CD4+ conventional T cells after HIPEC, with the magnitude of change strongly correlated with overall survival [93]. Understanding the impact of hyperthermia on peritoneal immune cells is crucial to harness its potential synergy with immunotherapy.

Taken together, current data does suggest a biological impact of heated IP treatment with transcriptomic and proteomic changes that may serve as a predictive or prognostic biomarker. However, much more rigor is required to reduce inter-study variance and identify putative pathways for prognostic or therapeutic studies. Additionally, both tissue- and treatment type-specific interactions with hyperthermia might exist, underscoring the need for more rigorous study. The potential benefits and adverse effects of hyperthermia are summarized in Table 2.

## 6. Biomechanical Effects of Hyperthermia

Biomechanical properties of solid tumors have a major role in the outcome of the therapy [98]. Drug penetration in tumor tissue during HIPEC is driven by convection (pressure gradient) and diffusion (concentration gradient) [58,99]. Depending on several properties including the drug, one mechanism can be dominant over the other. Tumors are characterized by an elevated interstitial fluid pressure (IFP), generating an outward flow (interstitial fluid flow; IFV) that opposes drug penetration [98]. Therefore, the depth of drug penetration is determined by the balance between inward diffusion and outward convection (Figure 4). Also, the combination of mechanical stress and fibroblast-derived TGF-β induces the transformation of fibroblasts to CAFs, which in turn produce collagen and other ECM molecules, leading to increased stromal stiffness, which hampers the infiltration of immune cells.

As noted above, diffusion is temperature-dependent, and a higher temperature is correlated with enhanced drug penetration. This was recently confirmed in a computational model of HIPEC [100].

Next to the effects of temperature on drug penetration, reduced matrix stiffness may facilitate the migration of immune cells in the tumor stroma. It has been observed that the permeability of tumor tissue decreases due to solid matrix heat treatment [101,102,103]. This may be due to the thermal transition of collagen. In acidic solution, reversible thermal transition occurs at 31–37 °C and is due to depolymerization of the smallest collagen fibrils. Irreversible thermal transition occurs between 37 °C and 55 °C, and is related to the complete defibrillation and unfolding of the native triple helical structure of collagen [104]. In a murine model of nanoparticle-mediated hyperthermia, the collagen within the tumor stroma changed from a mesh, fiber-like structure to a completely unstructured fibril [105]. It is well-documented that T-cell migration is hampered by increased stiffness and higher collagen density [106,107]. Also, the alignment of collagen fibers, which is generally perpendicular to the tumor surface, was shown to affect lymphocyte migration [108]. Other immune cell types are affected by stromal composition: collagen density was shown to modulate the immunosuppressive function of macrophages [109]. Taken together, these data suggest that hyperthermia may facilitate infiltration of immune cell populations by altered mechanical stromal properties.

## 7. Conclusions

Hyperthermia increases anticancer drug penetration and may promote infiltration by immune cells. However, the heat shock response elicited by HIPEC may prove to be double-edged, as enhanced expression of heat shock proteins might provoke unwanted, tumor-promoting effects next to reported antitumoral effects. Despite the increasing use of heated locoregional IP chemotherapy, there remains a lack of empirical and clinical evidence that clearly demonstrates the advantages of hyperthermia when treating peritoneal metastases and additional well-designed basic science; translational and clinical studies are urgently required to clarify the future treatment of PM.

## Figures and Tables

**Figure 1 cancers-15-04314-f001:**
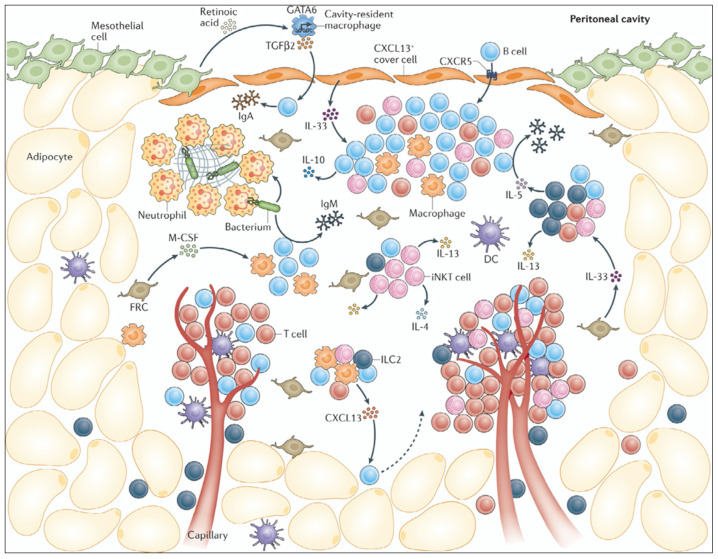
Immune environment in the healthy peritoneal cavity. Fat-associated lymphoid clusters (FALCs) are dense immunological structures found within the omentum. FALCs are capped by a layer of CXCL13 + FALC cover cells interspersed with the mesothelial cells. FALCs also contain a network of fibroblastic reticular cells (FRCs), which recruit and modulate immune cells. Mesothelial cells also release retinoic acid, which sustains GATA6-expressing cavity-resident macrophages in the peritoneum, which induce B-cell class-switching to IgA via transforming growth factor-β2 (TGFβ2). Neutrophils also are home to FALCs during acute inflammation and form neutrophil extracellular traps that capture contaminants or cancer cells. Reprinted with permission from [30].

**Figure 2 cancers-15-04314-f002:**
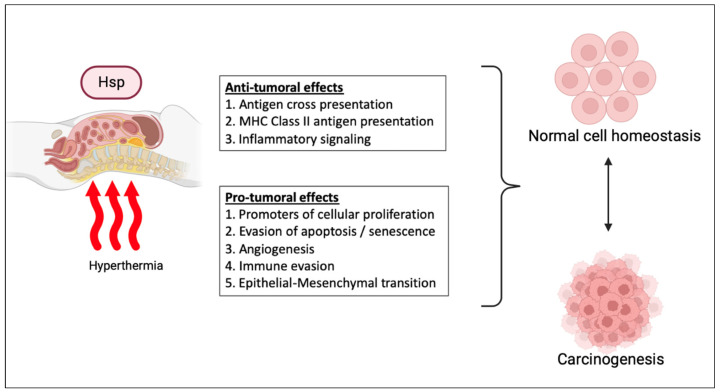
Effects of hyperthermia-related paracrine factors on carcinogenesis.

**Figure 3 cancers-15-04314-f003:**
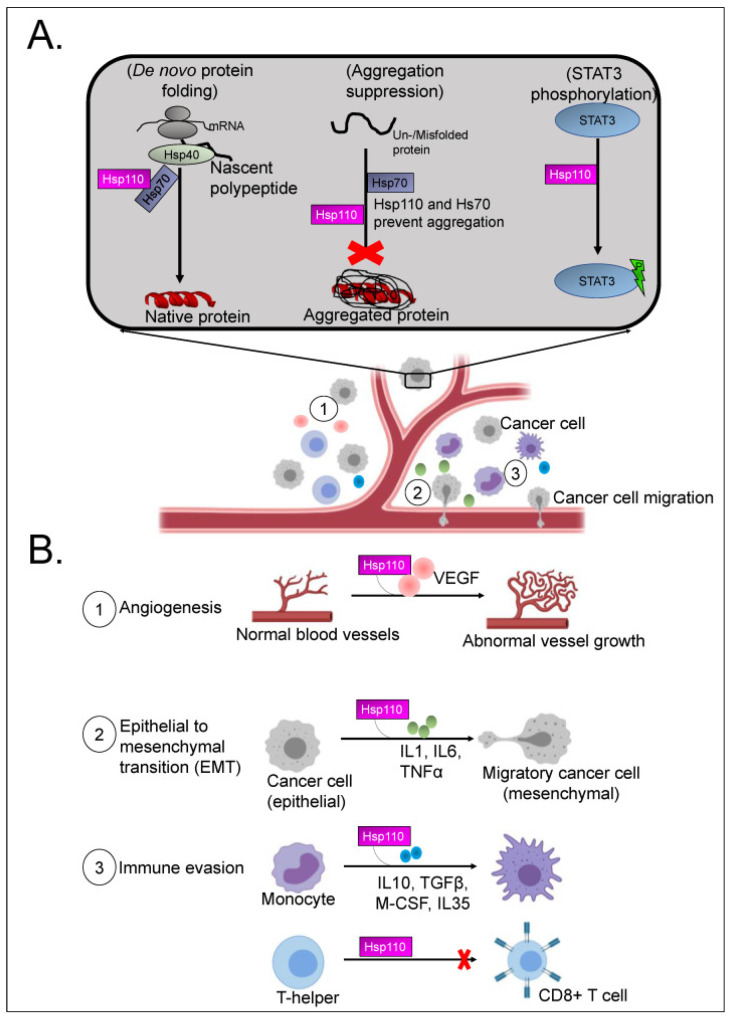
Mechanisms of carcinogenesis and pro-tumor effects of heat shock protein (HSP) 110. (**A**) Heat shock proteins have a chaperone function and lead to STAT3 phosphorylation. (**B**) Pro-tumoral effects of heat shock proteins (specifically HSP110) include angiogenesis, EMT, and immune evasion. Reprinted with permission from [79].

**Figure 4 cancers-15-04314-f004:**
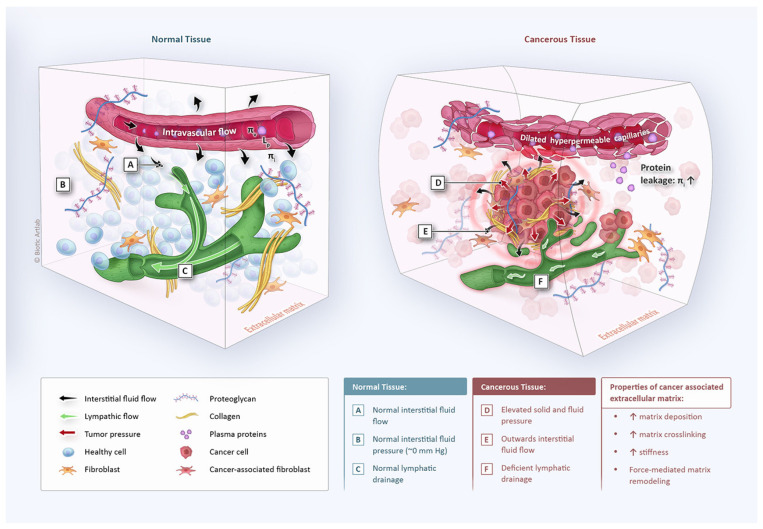
Key differences in tissue architecture and biophysics between normal and cancerous tissue. In normal tissue (**left**), fluid hemostasis is preserved through balancing fluid inflow and outflow, resulting in a low or even negative interstitial fluid pressure (IFP). Cancer tissue (**right**) is characterized by morphologically and functionally deficient blood and lymphatic microvessels, increased matrix deposition, and cancer cell proliferation. These changes result in increased solid and fluid pressure and elevated matrix stiffness, leading to a radially outward leakage of interstitial fluid. Reprinted with permission from [98].

**Table 1 cancers-15-04314-t001:** Summary of ongoing and completed clinical trials.

Completed and Ongoing Clinical Trials of Hyperthermia for the Treatment of PM				
Study	Cancer Type	Treatment	Status	Results
OVIP1 (NCT02567253)	Ovarian Cancer	Normothermic (37 °C) vs. Hyperthermic (41 °C) Cisplatin (H)IPEC	Completed	Hyperthermia increased the absorption rate of cisplatin by 16.3% [53]; clinical results not yet published.
NCT02739698	Ovarian Cancer	Normothermic (37 °C) vs. Hyperthermic (42–43 °C) Paclitaxel (H)IPEC	Completed	No difference observed in tissue penetration, pathological response, or apoptosis [8]
HyNOVA (ACTRN12621000269831)	Ovarian Cancer	Normothermic (37 °C) vs. Hyperthermic (42 °C) Cisplatin (H)IPEC	Ongoing	

**Table 2 cancers-15-04314-t002:** Potential benefits and adverse effects of hyperthermia.

Potential Benefits and Adverse Effects of Hyperthermia	
Benefits	Adverse Effects
Direct cytotoxic effects	Increased systemic uptake of chemotherapy
Synergism with some chemotherapeutic compounds	Heat shock response
Increased tissue penetration	Immunosuppressive effects
Immune-enhancing effects	Scald injury to the peritoneal surface
Increased blood flow and decreased interstitial fluid pressure → improved drug delivery	Increased bacterial translocation

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
