# Peer review of "Effects of Hyperthermia and Hyperthermic Intraperitoneal Chemoperfusion on the Peritoneal and Tumor Immune Contexture"

_cancers, 2023, doi:10.3390/cancers15174314_

Round 1
Reviewer 1 Report
As stated in the Title: "EFFECTS OF HYPERTHERMIA AND HYPERTHERMIC INTRAPERITONEAL CHEMOPERFUSION ON THE PERITONEAL AND TUMOR IMMUNE CONTEXTURE" the authors provide an extensive review based on 109 collected references.
ad1. My major concern is the deficient outline of methods to perform the review and the conclusions based on. To my opinion the Preferred Reporting Items for Systematic Reviews and Meta-Analysis (PRISMA) including a 27-item checklist should be used to improve transparency of this review. These items cover all aspects of the manuscript, including title, abstract, introduction, methods, results, discussion, and funding. To ensure a review is valuable to users, authors should prepare a transparent, complete, and accurate notification of why the review was done, what they did, and what they found.
ad2. The review is focussing HIPEC and most relevant references are related rather than to hyperthermia alone. The Title should delete the term hyperthermia and read : "EFFECTS OF HYPERTHERMIC INTRAPERITONEAL CHEMOPERFUSION ON THE PERITONEAL AND TUMOR IMMUNE CONTEXTURE"
ad3. line 385-386 :However, the heat shock response elicited by HIPEC may provoke unwanted, tumor promoting effects, specifically through enhanced expression of heat shock proteins. This statement in the conclusion is misleading. The role of intracellular chaperones and co-chaperones in sustaining transformation and cancer progression is well known, as the attitude of cancer cells to become addicted to HSP overexpression. However, it is widely accepted that HSPs are released by different types of cancers following hyperthermic therapy and may be associated with their plasma membranes. Several studies describe the tumour-specific immunogenicity of these released or surface-localized HSPs, a function associated with their ability to chaperone antigenic peptides and to activate anti-tumor innate and adaptive immunity. There is an imbalanced citation of mechanisms to this important topic and no clinical findings to support that HIPEC promotes cancer progression.
Author Response
ad1. My major concern is the deficient outline of methods to perform the review and the conclusions based on. To my opinion the Preferred Reporting Items for Systematic Reviews and Meta-Analysis (PRISMA) including a 27-item checklist should be used to improve transparency of this review. These items cover all aspects of the manuscript, including title, abstract, introduction, methods, results, discussion, and funding. To ensure a review is valuable to users, authors should prepare a transparent, complete, and accurate notification of why the review was done, what they did, and what they found.
Thank you for your suggestion. We agree that systematic reviews of clinical experiments and interventions should conform to the PRISMA guidelines in order to prevent biased clinical decision making. However, our work is a (narrative) review of selected basic and translational mechanisms related to the impact of hyperthermia on the immune system in peritoneal metastases. We do not aim at completeness, do not summarize clinical treatment effects, and do not intend to guide clinical decision making. Therefore, we consider that the PRISMA guidelines are not relevant for our work.
ad2. The review is focussing HIPEC and most relevant references are related rather than to hyperthermia alone. The Title should delete the term hyperthermia and read : "EFFECTS OF HYPERTHERMIC INTRAPERITONEAL CHEMOPERFUSION ON THE PERITONEAL AND TUMOR IMMUNE CONTEXTURE"
We respectfully disagree. There are to the best of our knowledge no RCTs investigating the effect of hyperthermia (without hyperthermic chemoperfusion) specifically on PM, and little clinical interest. Of course, due to the advent of HIPEC, the available literature is mostly comprised of investigations detailing this treatment modality. Little research has however strived to spotlight the role of hyperthermia specifically. We have clearly made this distinction in our manuscript, and strived to separate hyperthermia-specific effects, chemoperfusion-specific effects and their interactions.
Furthermore, much of the translational work done in this context makes no distinction between the source of hyperthermia, as the mechanisms involved are temperature-specific rather than treatment modality-specific. As such, we would opt to keep hyperthermia in the title, to clearly distinguish this work from other reviews which analyse HIPEC as a treatment modality (of which several have already been published), as this review is novel in giving a detailed summary of the effects of hyperthermia seen separately, and in conjunction with, intraperitoneal chemoperfusion.
ad3. line 385-386: However, the heat shock response elicited by HIPEC may provoke unwanted, tumor promoting effects, specifically through enhanced expression of heat shock proteins. This statement in the conclusion is misleading. The role of intracellular chaperones and co-chaperones in sustaining transformation and cancer progression is well known, as the attitude of cancer cells to become addicted to HSP overexpression. However, it is widely accepted that HSPs are released by different types of cancers following hyperthermic therapy and may be associated with their plasma membranes. Several studies describe the tumour-specific immunogenicity of these released or surface-localized HSPs, a function associated with their ability to chaperone antigenic peptides and to activate anti-tumor innate and adaptive immunity. There is an imbalanced citation of mechanisms to this important topic and no clinical findings to support that HIPEC promotes cancer progression.
We agree that both beneficial (such as antigen cross presentation mediated by HSP70/90) and pro-tumoral effects (such as those enacted by eHSPs on ECM remodeling and CAFs) are present and of note in this context, as detailed from line 201-234, and presented in figure 2. Our conclusion might not fully reflect this. We have rewritten our conclusion 385-386 to bring it more in line with the findings presented above: “However, the heat shock response elicited by HIPEC may prove to be double edged, as enhanced expression of heat shock proteins might provoke unwanted, tumor promoting effects next to reported antitumoral effects”.
Reviewer 2 Report
Well written manuscript, topic is of interest for the readers of Cancers. The relevant publications in the field were cited. The manuscript is scientifically sound. The conclusions are covered by the presented data.
All in all some improvements are necessary:
1. Simple summary is lacking! please add.
2. Citation style is heterogeneous (line 369) and uncommon for Cancers. Please use the following form all over the manuscript: This is the text and here is the right place for the reference [1].
3. References are not in the requested format - please change.
4. By using 3 reprints of previously published papers some questions of the originality of the content are raised. Please comment.
5. line 268: This statement is much to general. There are a lot of trials even randomized phase III-trials in the field of hyperthermia with clear clinical benefits. Please specify that you are speaking about hyperthermia in the context of intraperitoneal treatments.
Very well written manuscript, quality of English language is fine.
Some minor typos need to be fixed:
for example:
additional spaces in lines 52, 119, 141, 183, 208, 270
additional point in line 114
Author Response
1.Simple summary is lacking! please add.
If there is additional space for a simple language summary, we would be happy to provide one. Can the editors advise on this?
- Citation style is heterogeneous (line 369) and uncommon for Cancers. Please use the following form all over the manuscript: This is the text and here is the right place for the reference [1].
This has been amended.
- References are not in the requested format - please change.
The references have been reformatted using the MDPI Endnote style listed here: https://www.mdpi.com/journal/cancers/instructions#references
- By using 3 reprints of previously published papers some questions of the originality of the content are raised. Please comment.
Three figures have been reprinted:
- Figure 1 details the immune environment of the healthy peritoneal cavity.
- Figure 3 demonstrates the mechanisms of HSP 110
- Figure 4 summarizes biophysical properties between normal and tumor tissues.
All three figures provide necessary background information to further support our comprehensive review of mechanisms of hyperthermia in the context of peritoneal metastases. Furthermore, none of these figures cited detail hyperthermia in this context. A comprehensive review of this topic has not been undertaken before, and the concept of integrating both biophysical, pharmacological and immune effects of localized hyperthermia in the context of peritoneal metastases in detail is completely novel and original. As such, we firmly believe that this work, while supported by previously published work, represents a completely novel and original review manuscript.
- line 268: This statement is much to general. There are a lot of trials even randomized phase III-trials in the field of hyperthermia with clear clinical benefits. Please specify that you are speaking about hyperthermia in the context of intraperitoneal treatments.
We agree, and have made this distinction clear in the text.
- Some minor typos need to be fixed: for example: additional spaces in lines 52, 119, 141, 183, 208, 270 additional point in line 114
We have rectified this.
Reviewer 3 Report
Very well conducted and extensive literature review. A table should be added on clinical trials showing clear advantages of hyperthermia and also clear disadvantages. Can you make a table of NAT Gov Trial studies on this topic? I think it would be useful. To add also more discussion and a table on toxicities that in this work are poorly reported and poorly defined. These tables would add interest to the doctors concerned. I think that with these minor improvements the work will be more understandable to our readers and would rebalance the article too moved on laboratory data always conflicting and with conflicting conclusions with respect to clinical data and therapeutic results
The article is written in correct and relatively fluent English. But this is because it reports a lot of data that is not easy to write.
Author Response
- Very well conducted and extensive literature review. A table should be added on clinical trials showing clear advantages of hyperthermia and also clear disadvantages. Can you make a table of NAT Gov Trial studies on this topic? I think it would be useful.
Only three trials investigating either whole-body hyperthermia or the effect of hyperthermia in intraperitoneal chemotherapy are registered. We have added a table (Table 1) summarizing these studies, their status and these results.
- To add also more discussion and a table on toxicities that in this work are poorly reported and poorly defined. These tables would add interest to the doctors concerned.
Toxicities are largely dependent on type of chemotherapy, rather than hyperthermia alone. Furthermore, none of the current studies have demonstrated increased morbidity when comparing hyperthermic vs normothermic chemotherapy. However, due to only a small number of studies published, we feel that a detailed description of potential morbidities would be hypothetical at best. We have included a table (table 2) with proposed benefits and disadvantages of hyperthermia in the context of PM, which includes potential toxicity associated with hyperthermia and hyperthermic chemotherapy.
Round 2
Reviewer 1 Report
No further comments